# Nanoparticles as Emerging Labels in Electrochemical Immunosensors

**DOI:** 10.3390/s19235137

**Published:** 2019-11-23

**Authors:** Alba Iglesias-Mayor, Olaya Amor-Gutiérrez, Agustín Costa-García, Alfredo de la Escosura-Muñiz

**Affiliations:** NanoBioAnalysis Group-Department of Physical and Analytical Chemistry, University of Oviedo, 33006 Oviedo, Spain; iglesiasalba@uniovi.es (A.I.-M.); amorolaya@uniovi.es (O.A.-G.); costa@uniovi.es (A.C.-G.)

**Keywords:** nanoparticles, labels, electroactivity, electrocatalysis, electrochemistry, immunosensors, biosensors, nanomaterials, nanochannels

## Abstract

This review shows recent trends in the use of nanoparticles as labels for electrochemical immunosensing applications. Some general considerations on the principles of both the direct detection based on redox properties and indirect detection through electrocatalytic properties, before focusing on the applications for mainly proteins detection, are given. Emerging use as blocking tags in nanochannels-based immunosensing systems is also covered in this review. Finally, aspects related to the analytical performance of the developed devices together with prospects for future improvements and applications are discussed.

## 1. Introduction

Among electrochemical biosensors, immunosensors, based on the principle of high biospecific affinity interactions between antigens and the corresponding antibodies, have attracted considerable attention in many fields, such as clinical diagnosis or environmental monitoring. Electrochemical immunosensors are easy to use, but usually the affinity interaction generates a small current response that is not adequate to satisfy the desired sensitivity, being essential the development of efficient signal amplification strategies. Although enzyme-based labels are the most common, there are critical problems in the construction of enzymatic-based immunosensors such as the need for harsh conditions to keep the enzymatic activity. Additionally, natural enzymes have some intrinsic drawbacks, such as low thermal and environmental stability due to denaturation, sensitivity of catalytic activity to environmental conditions, non-conductivity and difficult and expensive synthesis and purification. The shortcomings associated with the use of enzymes as labels have promoted their substitution by nanomaterials that do not present these deficiencies and have many advantages over them.

Nanoparticles (NPs) have been extensively studied compared to bulk materials due to their unique features, including their catalytic, electric, optical and magnetic properties, being electrochemical analysis one of the fields with higher potential application [1]. Thanks to their excellent stability against denaturation, ease of mass production, ease of bioconjugation, and lower cost, many NP-based biosensors, mainly DNA sensors [2,3] and immunosensors, have drawn widespread attention in bio nanotechnology, offering benefits compared to traditional methods in terms of time of analysis, sensitivity and simplicity [4,5,6,7,8].

From the variety of immunosensors, those based on electrochemical detection are of special relevance, thanks to the inherent advantages of the electrochemistry in terms of sensitivity, simplicity, low cost of analysis and miniaturization possibilities, among others [9,10].

NPs have been extensively applied in electrochemical immunosensors as modifiers of electrode surfaces for improving both the electronic transference and the ability for antibody immobilization. Metallic NPs are the most widely used here, with carbon nanomaterials being also recently proposed for such purposes [11,12,13]. NPs also appear as emerging outstanding labels in electrochemical immunosensors, as alternative to classical enzymatic tags [14]. Most metallic NPs have redox properties allowing their direct sensitive detection. Moreover, many such NPs exhibit electrocatalytic activity towards different reactions, which has been approached for the development of ultrasensitive immunosensors. Finally, NPs can also be used as physical blocking agents in i.e. nanochannel-based electrochemical immunosensing systems.

Despite the increasing interest and number of published works on NPs as labels in electrochemical biosensing, there is a lack of recent reviews covering this field, mainly focusing on immunosensing. In this context, the aim of this review is to present a general overview about the most representative and recent applications of NPs as labels in electrochemical immunosensors, based on the aforementioned properties. Such immunosensors have been mostly applied for the detection of biomolecules with clinical interest, such as cancer biomarkers or tumour cells. The obtained results show that the developed technologies can be valid alternatives to the traditional methods and are currently the objective of extensive research.

## 2. Nanoparticles as Electroactive Labels

Some NPs possess redox properties that make them easy to be directly detected with electrochemical techniques, without the need for additional steps/reactions after the immunoreaction in which they act as labels. The most representative examples are detailed below.

### 2.1. Gold Nanoparticles (AuNPs)

Gold nanoparticles (AuNPs) stand out from the variety of NPs used as labels in immunosensing due to their simple synthesis, narrow size distribution, optical and electrochemical properties and easy bioconjugation alternatives. The advantageous properties of AuNP-based immuno and DNA electrochemical assays have been extensively exploited in the last few years [15,16]. The first works here were based on the NPs dissolving/destruction in aggressive acidic reagents, followed by the detection of the resulting Au (III) ions by anodic stripping voltammetry (ASV). One of the pioneer works following such strategy was reported by Limoges’s group, who also used the AuNPs as tags in an immunoassay for immunoglobulin G (IgG) detection at µg/mL levels [17].

After that, Liu and Lin [18] introduced the advantage of using magnetic particles as platforms of the immunoreaction for the same analyte determination, lowering the detection limits at ng/mL levels. In addition to the direct determination of IgG proteins, immunoassays for other analytes determination based on the ASV of AuNP tags have also been reported. For example, a disposable microfluidic device for the detection of *Salmonella typhimurium* through a magneto-immunoassay using both magnetic particles and AuNPs linked to specific antibodies (Figure 1A) was reported by de Oliveira et al. [19], reaching a limit of detection as low as 7 cells/mL. It also deserves to be highlighted the possibility of tagging AuNPs labels with different metal ions, for the multiplexing of different tumour markers [20].

Despite the high sensitivity of the ASV detection of the Au (III) resulting of the NP dissolving/destruction, the need of hazardous reagents in this process has limited its practical application in immunosensors as a reliable alternative to traditional immunoassays based on enzymatic labels. In this sense, Costa-García’s group was the pioneer in the development and application of an alternative methodology based on the direct detection of AuNPs without the need for previously dissolving them in highly acidic media [21]. The strategy is based on the electrochemical oxidation of the NPs surface by applying a low oxidative potential in diluted hydrochloric acid, followed by the electrochemical voltammetric reduction back of the Au (III) to Au (0). Later on, Merkoçi’s group combined this strategy with the labelling of antibodies with AuNPs and the advantages of using magnetic particle platforms, for the detection of IgG protein at pg/mL levels [22]. In 2011, de la Escosura-Muñiz et al. studied for the first time the effect of the size of AuNPs on the aforementioned direct electrochemical detection when used as electroactive labels in an immunoassay for IgG determination on magnetic particle platforms [23]. Their findings suggest a better performance for small NPs (5 nm AuNPs) instead of the standard Turkevich’s ones (20 nm AuNPs) due to their higher surface area, as illustrated in Figure 1B.

### 2.2. Silver Nanoparticles (AgNPs)

The excellent electroactivity of silver metal together with the well-defined sharp voltammetric peaks associated to the process of oxidation of Ag (0) to Ag (I) make silver nanoparticles (AgNPs) to be of great potential for analytical applications, as reviewed by Compton’s group [24]. Here, the presence of chloride ions in the electrolyte solution is of key relevance for forming the AgCl specie that facilitates the voltammetric oxidation.

Based on that principle, Ting et al. proposed the use of Ag tags in the development of an immunosensor for prostate specific antigen (PSA) detection at fg/mL levels [25]. In the same vein, an electrochemical biosensor for clenbuterol using melamine functionalized AgNPs was reported by Miao and co-workers [26] (Figure 2A), reaching limits of detection at pg/mL levels. Similarly, antibodies to tick-borne encephalitis virus (TBEV), one endemic flavivirus that can cause serious infections in humans, were detected at 50 IU/mL using this methodology [27].

The high susceptibility of Ag to oxidation makes easy its combination with Au to obtain bimetallic AuAgNPs [28] with such NPs having the benefits of both metals. In this sense, Merkoçi’s group reported first the synthesis and electrochemical characterization of AuAgNPs [29] (Figure 2B) and then, applied them for the quantification of *Escherichia coli* and *Salmonella typhimurium* bacteria, taking advantage of the affinity of Ag for cell surface macromolecules [30] (Figure 2C). These findings opened the way for the development of low-cost and quickly electrochemical detection of bacteria as alternatives to traditional culture-based methods.

### 2.3. Quantum Dots (QDs)

Quantum dots (QDs) are semiconductor NPs with spherical shape and a diameter between 1–12 nm. Nowadays, they are one of the most studied nanomaterials, mainly because of their unique optical and semiconductive properties. They were discovered in the early 1980s by Alexey Ekimov during his research on semiconductor nanocrystals [31,32]. Some of the novel characteristics of QDs are their narrow spectral bands, high photoluminescence emission quantum yields and size-tunable emission profiles. This, altogether, make them as excellent potential labels to be used in bioassays [33]. Apart from their optical properties, valuable information can be provided by their electrochemical behaviour, broadly studied by Bard’s group in 2005 [34]. QDs have normally a core@shell structure made of semiconductors. The one in the outer layer is used to protect the core against possible oxidation reactions that could release the inner ions. Different organic capping ligands are also used to control the solubility of QDs and their functionalization, [35] with the aim of using them in bioassays, especially in optical biosensing, because of their size-controlled luminescence [36].

In addition to their well-established optical properties/applications, QDs have also inherent electroactivity coming from their metallic components that make them easy to detect with electrochemical techniques. The typical strategy consists in the QDs dissolution in highly acidic/oxidative media followed by the ASV detection of the metal ions released. Moreover, the use of QDs made of different metals allows to do the simultaneous detection of different targets through the specific potential of re-oxidation of each metal. In this sense, Wang and co-workers were pioneers in the use of QDs of different metals (Pb, Zn, Cd) as labels for the simultaneous detection of different analytes through ASV analysis [37].

ASV detection after QDs labels acidic dissolving has also extensively used in immunosensing in the last decade. As example, PSA, a biomarker for prostate cancer, has been detected using CdS QDs as labels in a sandwich-type immunoassay at clinically relevant levels of pg/mL [38].

Other QDs, CdSe QDs, have been combined with zirconia NPs (ZrO_2_) by Lu et al. [39] in the development of a highly selective electrochemical immunosensor for the detection of organophosphorylated butyrylcholinesterase (OP-BChE), a biomarker of the exposure to toxic organophosphorus agents, at environmental relevant levels of ng/mL. Core@shell CdSe@ZnS QDs, with ASV detection of released Cd, have also been highly used in electrochemical immunosensing. For example, Martín-Yerga et al. proposed the ASV detection of CdSe@ZnS QDs tags in electrode arrays [40], later applied for the detection of celiac disease biomarkers at clinical relevant levels (around 2 U/mL) [41,42] (Figure 3A). In the same vein, Pinwattana et al. used CdSe@ZnS QDs for the quantification of phosphorylated bovine serum albumin (BSA-OP) at the ng/mL scale [43].

As in the case of the AuNPs, the QDs detection by stripping after dissolving is practically limited by the need of hazardous reagents for the NP dissolution and the metal ions release. In this context, Merkoçi’s group was the first in proposing the direct detection of CdS QDs, based on the reduction/re-oxidation of Cd (II) in the surface of the NP, without the need of decompose the QDs [44]. This methodology was later applied in an immunosensor using CdSe@ZnS QDs tags for the detection of apolipoprotein E (ApoE), an Alzheimer’s disease biomarker, at clinical relevant levels of ng/mL [45] (Figure 3B). The same authors proposed later a signal amplification strategy based on the use of bismuth-modified electrodes for improving the Cd detection (Figure 3C), which was applied for the determination of human IgG (HIgG) at ng/mL levels in a model immunoassay [46].

### 2.4. Other Nanoparticles (NPs)

Some other nanoparticles with outstanding electrochemical properties, but not yet been used as labels in biosensors, deserve to be briefly included in this review due to their great potential for such application. That is the case of cerium oxide nanoparticles (CeO_2_ NPs) and mercury selenide nanoparticles (HgSe NPs). Copper-based metal nanoparticles (CuNPs), that have been proposed only for DNA hybridization biosensing, are also listed in this section.

#### 2.4.1. Cerium Oxide Nanoparticles (CeO_2_ NPs)

Among all the metal oxide-based nanoparticles, cerium oxide nanoparticles or nanoceria have attracted significant attention owing to their singular properties, especially as catalysts. Their crystal structure have lots of oxygen vacancy defects, and because of that, they exhibit a very important oxygen storage capacity [47]. For this reason, the oxidation state of cerium at the NP surface can vary easily between +3 and +4, so they can both act as oxidizing and reducing agents [48]. This allows them, apart from acting as catalysts, to mimic the activity of enzymes in biosensors [49]. CeO_2_ NPs have been widely used in biosensing based on such mimetic properties. Ispas and colleagues have investigated the electrochemical behaviour of nanoceria towards the oxidation and reduction of hydrogen peroxide, resulting in a highly sensitive H_2_O_2_ detection technique with very low response times [50] opening the pathway to CeO_2_ NPs for their use in biosensing applications. On this basis, Chaturvedi et al. have developed a CeO_2_ NPs-Pt-graphene nanocomposite for the detection of glucose and xanthine, adding peroxide-producing oxidase or superoxide-producing oxidase, respectively [51].

With the aim of avoiding the use of enzymes, this peroxidase mimetic activity of CeO_2_ NPs has been combined with the catalytic properties of AuNPs to oxidize glucose into gluconate and hydrogen peroxide [52] which was applied for developing a non-enzymatic glucose biosensor with good analytical characteristics [53].

Recently, it has also been developed a novel method for the quantification of CeO_2_ NPs based on their oxidative effect towards ferrocyanide redox system, with great potential for its further application in biosensors [54].

#### 2.4.2. Mercury Selenide Nanoparticles (HgSe NPs)

Mercury selenide (HgSe) is a very interesting material characterized by its high electron mobility and large electron concentration, extensively investigated in the area of optoelectronics [55]. The electrochemical behaviour of HgSe has been studied in mercury electrodes since the early 1960s, concluding that selenious acid is irreversibly reduced to HgSe in an acidic media. The cathodic stripping of HgSe film has been later used in the quantification of selenium [56]. After that, different studies on the optimization of HgSe thin films by electrochemical atomic layer epitaxy have been reported, with the different films being grown layer by layer making use of surface limited reactions such as under potential deposition (UPD). UPD is a highly interesting process consisting on the deposition of atomic layers on another element at a different potential of the one needed for the deposition on the element on itself [57].

More recently, it has been demonstrated that selenium and mercury have a toxicological antagonism in animals, occurring normally bioaccumulation of both elements [58]. Such co-accumulation may be due to the existence of HgSe NPs in the liver of cetaceans, assuming that they are the final metabolic product of the lifesaving mechanism in different biological systems [59]. In this context, engineered water-stabilized HgSe NPs have been synthesized and characterized by Bouzas-Ramos and co-workers [60]. Using these NPs and exploiting the ability that electrochemical techniques have to pre-concentrate different metals on the surface of the electrode, a rapid, simple and sensitive quantification of HgSe NPs has been carried out for the first time by Iglesias-Mayor et al. [61] In this work, HgSe NPs were quantified within two orders of magnitude, obtaining good reproducibility, repeatability and limit of detection, showing great potential for further application as tags in electrochemical immunosensors.

#### 2.4.3. Copper-Based Nanoparticles (CuNPs)

Copper-based metal nanoparticles are attracting attention in bioanalysis due to their biocompatibility, low toxicity and outstanding optical properties. Some recent approaches have taken advantage of the in situ generation of CuNPs after DNA amplification, followed by NP dissolving and Cu ions detection by stripping voltammetry. This strategy has been combined with the use of aptamers for the detection of PSA biomarker at fg/mL levels [62]. The strong interaction between glutathione (GSH) and copper ions [63] has also been approached for the detection of GSH after formation of DNA-templated CuNPs and later voltammetric detection, as described above [64]. This analytical signal readout has also been used in the development of a method for quantifying endonuclease activity [65].

On the other hand, DNA-templated copper NPs are also considered as functional probes in bioanalysis [66]. They are synthesized due to the clustering of Cu onto DNA scaffolds [67] in a fast and efficient way [68]. Aptasensors for the detection of microRNA have also been reported using CuNPs, for example, Wang et al. built a biosensor for microRNA 21 based on the combination of the electroactivity of CuNPs and different amplification strategies [69], with an ultra-low limit of detection at levels of ag/mL, and having reliable results in the analysis of real blood samples. A simpler biosensor for microRNA which only uses exonuclease and copper nanoparticles has been reported by Miao and co-workers [70], with high potential for clinical diagnosis.

## 3. Nanoparticles as Electrocatalytic Labels

Catalysis is one of the main fields of nanoscience and nanotechnology, with the development of highly catalytic and electrochemically stable catalysts being a crucial goal in electrochemical applications. The special surface characteristics of the NPs make them ideal candidates for their use in electrocatalytic processes, mainly due to the higher proportion of atoms at the NP surface than in bulk state, that results in a high surface to volume ratio.

Silver electrodeposition, hydrogen evolution reaction (HER), water oxidation reaction (WOR) and oxygen reduction reaction (ORR) are simple reactions on which the electrocatalytic activity of some NPs has been approached for electroanalytical applications.

Moreover, various nanomaterials have been found to mimic the activities of natural enzymes such as peroxidase, superoxide dismutase, catalase and oxidase, being thus termed as nanozymes [71,72]. Particularly, nanomaterials with peroxidase-like activity have been extensively used as alternatives to the enzyme horseradish peroxidase (HRP), the most popular label in enzyme-based immunosensors. 

The most representative/relevant works on the immunosensing application of the above mentioned electrocatalytic properties of NPs used as labels are described in this section.

### 3.1. Gold Nanoparticles (AuNPs)

Gold nanoparticles have grabbed considerable attention in last years because of their unique catalytic properties. Although bulk gold is an inert material towards redox processes, AuNPs have shown a different behaviour due to the quantum effects related with shape and size of the NPs. AuNPs have been found to exert an excellent electrocatalytic activity towards a variety of chemical reactions, and their ease of conjugation with different biomolecules make them ideal candidates for using as labels in the electrochemical detection of different proteins [73]. 

#### 3.1.1. Silver Electrodeposition

AuNPs have shown catalytic properties towards the electroreduction (advantageous alternative to chemical reduction) of Ag (I) to Ag (0), since the presence of NPs shifts the Ag reduction potential to less negative values. A selective electrodeposition of silver only onto the AuNPs surface was achieved by applying an adequate electrodeposition potential, being then the silver re-oxidized scanning to positive potentials, generating a voltammetric peak of currents proportional to the amount of AuNPs [74]. This sensitive AuNPs induced-electrocatalytic silver reduction in combination with the use of magnetic particle platforms was exploited for the development of an immunoassay for the quantification of HIgG as model protein at ultralow levels (fg/mL) (Figure 4A). A similar strategy was later reported for the detection of carcinoembryonic antigen (CEA) and alpha-fetoprotein (AFP) at pg/mL levels [75] (Figure 4B). The same analytes were further quantified at fg/mL by Yan et al., using streptavidin/nanogold/carbon nanohorn (SA/Au/CNH) as a novel signal tag to induce silver enhancement for signal amplification [76].

However, the practical implementation of this strategy is limited because of its incompatibility with screen-printed electrodes (SPEs), probably the most popular electrodes for point-of-care (POC) electrochemical immunosensing. The presence of silver in the Ag/AgCl printed pseudoreference electrode interferes in the detection of the selectively electrodeposited silver, making its discrimination difficult. Alternative methodologies compatible with SPEs, mainly the one based on the electrocatalytic hydrogen evolution, were later proposed.

#### 3.1.2. Hydrogen Evolution Reaction (HER)

The electrocatalytic effect of AuNPs towards the hydrogen evolution reaction (HER), that is the hydrogen ions (H^+^) reduction from a slightly acidic medium to form hydrogen (H_2_), has also been proposed as an outstanding methodology for the sensitive AuNPs detection on SPEs [77]. The electrocatalytic detection relies on the shift of the H^+^ ions reduction potential to less negative values in the presence of AuNPs.

De la Escosura-Muñiz et al. studied and optimized an AuNPs quantification methodology based on chronoamperometric measurements. By applying a constant voltage (typically −1V) the H^+^ ions reduction selectively occurs only in presence of AuNPs, so the cathodic current recorded at a fixed time is related to the amount of AuNPs. This methodology was first proposed for the detection of tumour cells cultured on SPEs through specific antibodies labelled with AuNPs, determining 4000 tumour cells per 700 µL of suspension [78] (Figure 5A)**.** Later on, a similar methodology was applied for the detection of circulating tumour cells (CTCs) [79] also taking advantage of the use of magnetic particle platforms for capturing/pre-concentrate the CTCs on the SPE surface, allowing to determine around 5 × 10^3^ cells/mL [80] (Figure 5B).

The use of magnetic particle platforms in combination with AuNPs tags and the electrocatalyic HER-based reaction was extensively exploited by the same group for the immunodetection of different analytes. As examples of protein biomarkers, they proposed the detection of HIgG at ng/mL levels [81] (Figure 5C), anti-hepatitis B antibodies at levels low enough for assuring the efficiency of vaccination (3 mIU/mL) [82] (Figure 5D) and Alzheimer’s disease biomarkers (ApoE and beta amyloid) at diagnostically relevant levels [83]. Furthermore, the same principles were approached for the quick detection and quantification of *Escherichia coli* O157:H7 in meat (minced beef) and water samples, without the need of broth enrichment, achieving detection limits in the order of 300–500 CFU/mL [84]. The method was also extended to DNA quantification, being applied for the isothermal amplification of *Leishmania* DNA using primers labelled with AuNPs [85].

#### 3.1.3. H_2_O_2_ Reduction

The peroxidase-like activity of various nanomaterials, including AuNPs [86], has attracted great attention in the last years. As an example, Jia et al. developed a sandwich-type immunosensor for the detection of squamous cell carcinoma antigen (SCC-Ag), a biomarker of cervical cancer, at fg/mL levels using Na-montmorillonite-polyaniline-AuNPs (Na-Mont-PANI-AuNPs) nanocomposites that exhibit excellent catalytic performance towards H_2_O_2_ reduction [87] (Figure 6A). In the same line, a competitive amperometric immunosensor was recently proposed for the sensitive detection of tetrabromobisphenol A bis(2-hydroxyethyl) ether (TBBPA-DHEE), a potential neurotoxin in environmental waters, using catalase functionalized AuNPs-loaded self-assembled polymer nanospheres (PS@PEI@CAT@AuNPs) nanocomposite. A low detection limit in the pg/mL scale (7 times lower than that obtained by enzyme-linked immunosorbent assay (ELISA)) was achieved due to the synergistic effect of AuNPs and catalase on the electrocatalytic H_2_O_2_ reduction [88].

### 3.2. Silver-Based Nanoparticles (AgNPs)

Silver nanomaterials, particularly AgNPs, have also attracted increasing attention from the electrocatalytic perspective, due to their quantum characteristics including large specific surface area and small granule diameter, showing also catalytic activity towards the reduction of H_2_O_2_. An immunosensor based on such electrocatalytic property was developed by Zhang et al. for the ultrasensitive detection of CEA, a broad spectrum tumour marker, at very low levels (fg/mL) [89]. The chronoamperometric detection of H_2_O_2_ reduction relied on the synergistic effect towards that reaction of Hemin and AgNPs from AgNPs@CS-Hemin/rGO nanocomposite (microporous carbon spheres loading silver nanoparticles-spaced Hemin/reduced graphene oxide porous nanocomposite) used as label (see Figure 6B). A hybrid nanocomposite made of Fe_3_O_4_ nanospheres and Ag@Au nanorods (Ag@Au-Fe_3_O_4_) was also proposed as electrocatalysts of H_2_O_2_ reduction for the detection of HIgG at fg/mL levels [92].

### 3.3. Platinum-Based Nanoparticles (PtNPs)

Noble metal nanoparticles, especially platinum (PtNPs) and palladium (PdNPs), have attracted extensive attention thanks to their superior biocompatibility, favouring the immobilization of antibodies, and their remarkable electrocatalytic activity towards H_2_O_2_ reduction [93,94]. Various sandwich-type electrochemical immunosensors based on the electrocatalytic performance of PtNPs towards H_2_O_2_ reduction have been reported for the quantitative determination of different biomarkers. Li et al. achieved the detection of hepatitis B surface antigen (HBsAg) at fg/mL levels by using a molybdenum disulphide@cuprous oxide–platinum (MoS_2_@Cu_2_O-PtNPs) nanocomposite as signal amplification label [90] (Figure 6C). Alternatively, Pei et al. also accomplished the sensitive detection of HBsAg at the same levels by using a composite based on Rh core and Pt shell nanodendrites loaded onto functionalized graphene nanosheets (Rh@Pt NDs/NH_2_-GS) as label [95]. Both immunosensors provided good potential for HBsAg detection in clinical diagnosis, with their detection limits being below the threshold value of serum HBsAg. 

A chronoamperometric immunosensor for PSA determination was developed by Feng et al. [91] using PtCu@rGO/g-C_3_N_4_ nanocomposite (PtCu bimetallic hybrid loaded on 2D/2D reduced graphene oxide/graphitic carbon nitride) as label through the enhanced catalysis of H_2_O_2_ reduction (Figure 6D). The fg/mL levels detected evidence its potential application in clinical monitoring of such prostate cancer biomarker. Based on the same principles, the quantitative monitoring of another clinically relevant protein tumour biomarker, AFP, at fg/mL levels was also proposed using PtNPs anchored on cobalt oxide/graphene nanosheets (PtNPs/Co_3_O_4_/graphene) as electrocatalytic labels [96].

### 3.4. Palladium-Based Nanoparticles (PdNPs)

Pd-based nanocomposites have also been proposed as excellent labels for immunosensing, thanks to their electrocatalytic properties towards the H_2_O_2_ reduction. The most representative examples found in the bibliography are detailed in this section.

As example, an ultrasensitive immunosensor for the quantitative detection of AFP cancer biomarker (fg/mL levels) using a nanocomposite made of graphene oxide and CeO_2_ mesoporous nanocomposite functionalized by 3-aminopropyltriethoxysilane supported Pd octahedral NPs (Pd/APTES-M-CeO_2_-GS), that exhibits an important electrocatalytic activity towards H_2_O_2_ reduction, was reported by Wei et al. [97]. Another protein of clinical relevance for cancer screening, PSA, was sensitively quantified at the same levels using Pd NPs loaded electroactive amino-zeolitic imidazolate framework-67 (Pd/NH_2_-ZIF-67) nanocomposite, taking advantage of the synergistic catalytic effect of PdNPs and the electroactive NH_2_-ZIF-67 to decompose H_2_O_2_ [98] (Figure 7A).

Palladium-doped graphitic carbon nitride nanosheets (g-C_3_N_4_-PdNPs) were also found to have outstanding intrinsic peroxidase-like activity [99]. This nanocomposite was used as label for the detection of saxitoxin (STX), a relevant paralytic shellfish toxin at pg/mL levels [100] (Figure 7B).

### 3.5. Multimetallic NPs

Multimetallic nanomaterials have raised considerable concern in worldwide research because of their novel chemical and physical properties derived from the synergistic and electronic effects between the metals, coupled with the specific nanostructures, making them promising for catalytic applications [101]. These features are due to the change of atomistic arrangement and electronic states of the different metals in the nanomaterial due to the differences of electronegativity and electron configurations compared to single metals [102]. Additionally, in the case of noble metals NPs, their alloying with less-expensive non-noble metals (Cu, Fe, Co, Ni etc.) forming multimetallic NPs is interesting not only for being an effective approach to promote catalytic activity, but also from an economic perspective [103].

In general, bimetallic or multimetallic nanomaterials display highly enhanced electrocatalytic performance as compared to single metal catalysts for electrochemical applications. In the case of their use as labels in electrochemical immunosensors, the most exploited electrocatalytic property relies in their peroxidase-like activity towards the H_2_O_2_ reduction. Representative examples of such application are commented on below. 

#### 3.5.1. Gold-Based Multimetallic Nanoparticles

SCC-Ag cancer biomarker has been detected at clinically relevant levels of fg/mL through immunoassays, using different multimetallic NPs with catalytic activity towards the H_2_O_2_ reduction. As representative examples, Au@Ag/Au NPs core@double shell NPs [104] and amino functionalized cobaltosic oxide@ceric dioxide nanocubes combined with gold@platinum NPs (Co_3_O_4_@CeO_2_-Au@Pt) [105] have been proposed with such a purpose. 

Gold@silver core@shell nanoparticles and disordered cuprous oxide nanocomposites (Au@Ag-Cu_2_O), with an excellent catalytic activity due to the synergy between Au and Ag and Cu were successfully prepared and used as label for amplified PSA detection at fg/mL levels [106]. Also based on core@shell gold@silver NPs, in this case loaded by polydopamine functionalized phenolic resin microporous carbon spheres (Au@Ag/PDA-PR-MCS), an electrochemical immunosensor was devised for the detection of AFP at the same levels [107]. The high electrocatalytic activity of Au@Ag/PDA-PR-MCS towards the H_2_O_2_ reduction (Figure 8A) relied on the synergistic effect of Au@Ag NPs in combination with the relatively large surface area of PDA-PR-MCS that leads to a larger electrochemically active surface area. 

Gold@platinum core@shell NPs modified with neutral red functionalized reduced graphene oxide (rGO-NR Au@Pt) nanocomposite also exhibited high electrocatalytic activity that was applied for the immunoelectrochemical determination of *Escherichia coli* O157:H7 at 90 CFU/mL levels [108].

A nanocomposite made of cubic Au@Pt NPs dendritic nanomaterial functionalized with nitrogen-doped graphene loaded with copper ions (Au@Pt DNs/NG/ Cu^2+^) was used as label for the early detection of CEA at levels low enough for diagnostic applications (fg/mL) [109] (Figure 8B). 

The combination of Au with other noble metal such as Pd has also been used in nanocomposites for its peroxidase-like activity in immunosensing. As example, gold@palladium nanoparticles loaded with molybdenum disulfide functionalized multiwalled carbon nanotubes (Au@Pd/MoS_2_@MWCNTs) have been proposed by Gao et al. for the detection of hepatitis B e antigen (HBeAg) at levels of clinical interest (fg/mL) [110]. A particular case is the one reported by Pei et al. in which the signal amplification label, gold@palladium nanodendrites loaded on ferrous-chitosan functionalized polypyrrole nanotubes (Au@Pd NDs/Fe^2+^-CS/PPy NTs) not only showed electrocatalytic activity towards H_2_O_2_ reduction, but also was used as electroactive compound to directly amplify the current signal of Fe^2+^, without need of any additional substance [111]. Benefitting from both possibilities, amperometry and square wave voltammetry (SWV) were used, respectively, for the electrochemical detection of CEA at fg/mL levels (Figure 8C).

#### 3.5.2. Palladium–Platinum-Based Bimetallic Nanoparticles

In a lower extent, bimetallic Pd–Pt NPs have also been used as electrocatalytic labels in immunosensors, based on the above detailed principles. As representative example, Wang’s group reported two novel signal amplification strategies based on Pd–Pt NPs for the quantitative monitoring of PSA. In their first work a novel mesoporous core@shell palladium@platinum NP loaded by amino group functionalized graphene (M-Pd@Pt/NH_2_-GS) nanocomposite was used as label [112]. Later on, they reported the use of trimetallic PdPtCu nanospheres (m-PdPtCu) for the same purpose [113]. In both cases they detected PSA at fg/mL levels, having promising potential for clinical diagnosis. 

### 3.6. Iridium-Based Nanoparticles (IrNPs)

Iridium nanoparticles (IrNPs) have been considered as an excellent catalyst towards H_2_O_2_ reduction due to their selectivity and stability, large surface area and high density of active sites. That electrocatalytic property of IrNPs has been approached in a immunosensor for CEA detection at diagnostic threshold levels of fg/mL [114]. 

Even more interesting is the fact that the oxide form of the IrNPs exhibits catalytic ability towards the water oxidation reaction, as has been demonstrated under photochemical conditions [115]. Iridium oxide nanoparticles (IrO_2_ NPs) are interesting for biosensing thanks to their biocompatibility, catalytic activity and outstanding stability. A benefit towards other NPs is their ease of being electrocatalytically detected at neutral pH through the water oxidation reaction (WOR), which favours their direct detection in the same medium where immunoreactions usually occurs, avoiding the use of additional reagents. 

A representative work of such application is the one reported by Merkoçi’s group, in which the electrocatalytic activity of IrO_2_ NPs towards the WOR in neutral media was studied by cyclic voltammetry and chronoamperometry [116]. Fixing an adequate oxidative potential, the intensity of the current recorded in chronoamperometric mode is related with the amount of IrO_2_ NPs on the SPE transducer (Figure 9A). Combining the use of magnetic bead platforms and IrO_2_ NPs as labels of antibodies, they applied this catalytic method for the detection of ApoE, an Alzheimer’s disease biomarker at ng/mL levels (Figure 9B). 

Based on the same principles, an immunoassay for the detection of a flame retardant compound (BDE-47, a polybrominated diphenyl ether (PBDE)), was carried out by the same group, [117] reaching detection limits of ppb, that fulfil environmental control requirements (Figure 9C).

### 3.7. Ferromagnetic Nanoparticles

Ferromagnetic nanoparticles, mainly made of magnetite (Fe_3_O_4_ NPs) are quite popular in biological analysis due to their peroxidase-mimic activity, having advantages such as easier production, lower cost of synthesis, higher stability in harsh conditions and flexible storage conditions [118,119].

Li et al. [120] took advantage of Fe_3_O_4_ NPs for the development of a photoelectrochemical immunoassay for detecting PSA. Histidine-modified Fe_3_O_4_ (his-Fe_3_O_4_) was linked with the corresponding secondary antibody to form his-Fe_3_O_4_@Ab_2_. This conjugate acted as peroxidase-like reagent, catalysing the H_2_O_2_ reduction and therefore the simultaneous oxidation of 4-chloro-1-naphthol (4-CN) present in the media to benzo-4-chlorohexadienone, that precipitates on the sensor surface causing the blocking of the electron transfer. Hereby the target antigen was quantitatively detected via the decrease in the photocurrent signal, produced by the induced catalytic precipitation (Figure 10A). With such photoelectrochemical immunoassay, low detection limits in the fg/mL scale were obtained for the PSA biomarker.

### 3.8. Molybdenum Disulphide Nanoparticles (MoS_2_ NPs)

Molybdenum disulphide nanoparticles (MoS_2_ NPs) have been found to display electrocatalytic activity towards the hydrogen evolution reaction (HER) and towards the oxygen reduction reaction (ORR) [121].

In spite of their potential, as far as we know, only one example of application in immunoelectroanalysis is found in the bibliography. That is the work of Bouša et al. [122] who described the use of MoS_2_ NP conjugated with antibodies as signal-enhancing labels based on the chronoamperometric HER monitoring for rabbit IgG quantification at pg/mL levels (Figure 10B).

## 4. Nanoparticles as Blocking Agents in Nanochannels-Based Immunosensors

Nanoparticles size in the nanometric scale has also been approached for their use as physical elements for blocking/closing nanostructures, which was applied for immunoelectroanalysis when used as labels of antibodies. In this context, novel methodologies based on the blocking effect of AuNPs on the diffusion of electroactive species inside nanochannels-based platforms have been recently reported for biomolecules detection. 

Nanoporous alumina membranes stand out from the variety of nanoporous materials used for immunosensing [123,124]. Their high pore density and small pore diameter result in an easily functionalized substrate with high surface area. In this context, the capability of current tuning of such nanopore/nanochannel-based platform upon bioblocking using SPEs transducers and simple voltammetric detection mode has been proposed for the detection of proteins and DNA. The formation of the DNA duplex or the immunocomplex partially blocks the diffusion of electroactive species through the nanochannels, allowing researchers to detect and quantify the bioanalyte (Figure 11).

Such a methodology was approached for the label-free electrochemical detection of HIgG at µg/mL levels [126,127] and parathyroid hormone-related protein (PTHrP) cancer biomarker at clinical interest levels (ng/mL) [128], representing a simple biodetection alternative that was then extended to other immuno and DNA detection systems [129].

The sensitivity of the label-free assay was greatly improved later using AuNP tags as blocking agents in sandwich immunoassays. The presence of the AuNPs inside the nanochannels attached through the specific immunoreactions increases the blockage in the nanochannels, with this effect being strongly dependent on the NPs size. The nanochannels blockage was also enhanced by silver catalytic deposition, forming big silver shells that further decrease the diffusion of the redox indicator through the nanochannel, allowing to improve the detection limits up to ng/mL levels of HIgG [125]. In the same work, the ability of the nanochannels to act also as filters of big molecules was approached for the detection of a cancer biomarker in whole human blood samples. The micrometric components of the blood (red and white blood cells, platelets, and crystal salts) remain out of the membrane, while the cancer biomarker enters through the channels and is captured by the specific antibody. The further nanochannel blocking using AuNP tags and silver deposition allows to detect CA15-3 glycoprotein, a breast cancer biomarker at clinical relevant levels of 50 U/mL directly in human blood. A similar methodology was also applied for the detection of thrombin in whole blood at very low levels in of few ng/mL, which are within the range of clinical interest for the diagnostic of coagulation abnormalities as well as pulmonary metastasis [130] (Figure 12).

It is also worth highlighting the potential of these sensing systems for the in-situ monitoring of cell/bacteria secreted proteins, of high interest for the screening of anti-cancer drugs as well as for testing of novel antimicrobial/antivirulence agents [131,132].

## 5. Conclusions and Perspectives

Nanoparticles are emerging materials with outstanding potential for their use as labels in electrical immunosensing. Gold, silver, palladium and platinum are the main components of such particles, thanks to their direct electroactivity (redox properties) and/or their electrocatalytic activity towards secondary reactions. While the direct detection is faster and simpler, the electrocatalytic strategies are generally more sensitive, so the choice of nanoparticle and detection route is strongly dependent on the requirements of the concrete application. Attending to the electrocatalytic routes, those based on the hydrogen evolution reaction (HER) and the H_2_O_2_ reduction are most widely exploited, thanks to the variety of nanoparticles and nanocomposites able to electrocatalyse such reactions. However, of special relevance are recent works exploiting the ability of nanoparticles to electrocatalyse simple reactions such as the water oxidation reaction (WOR) and the oxygen reduction reaction (ORR), which occur at neutral pH without the need of additional reagents after the immunoassay. This is of special relevance for the building of highly integrated biosensing systems.

It is also worth highlighting the recent use of nanoparticles in combination with nanochannels for immunosensing approaches, with this being especially advantageous for real sample analysis, thanks to the filtering abilities of the nanoporous-based platforms.

Protein biomarkers of a variety of diseases, including tumour cells, are the target analytes on which such electrochemical immunosensors have been mostly applied. In most cases, pg/mL levels of interest for diagnostic applications are detected, being thus the sensitivity a challenge that has been overcome. Table 1 summarizes the NPs, strategies, analytes and main results reported over the last five years.

However, some important issues need to be solved for the definitive implantation of such sensing systems for routine analysis. Efficiency and long-term stability of the nanoparticle-antibody conjugates is a crucial issue that is not usually addressed in most of the reviewed works. Moreover, looking at the application as an alternative to well-established ELISA assays, multidetection abilities, either through electrode arrays or by simultaneously using nanoparticles with different properties should be strongly required. Efforts in this sense should be the next at the current state of the art.

Nevertheless, nanoparticle-based electrochemical immunosensors have the greatest potential in the field of the point-of-care (POC) analysis, in our opinion. The combination of the miniaturized electrochemical transducers, the cheap and portable electrochemical instruments and the well-studied properties of the nanoparticles make altogether ideal for POC applications. In this case, the most challenging and less reported issue is related to the sampling/introduction of the analytes in the immunosensor as well as the washing, etc. steps required, which difficult to have highly integrated real POC systems. Here the combination with microfluidics seems to be crucial. Considerable effort in this sense is expected in the next few years.

## Figures and Tables

**Figure 1 sensors-19-05137-f001:**
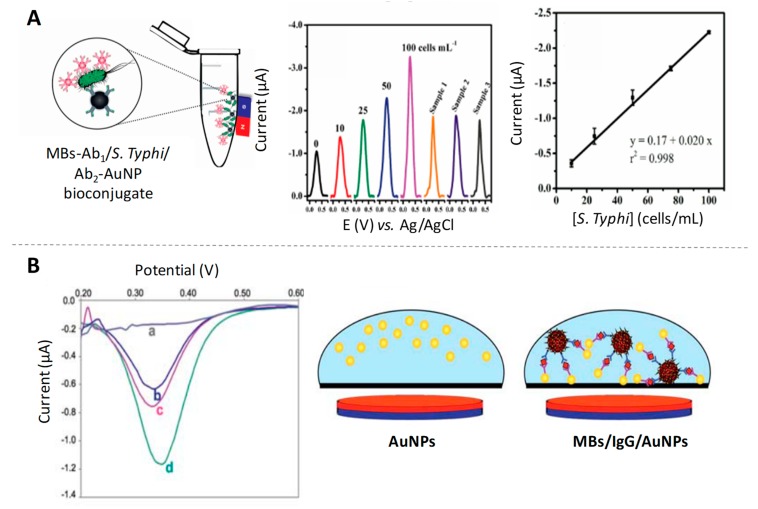
Gold nanoparticles (AuNPs) as electroactive labels. (**A**) Scheme of the magneto-immunoconjugate for the detection of *Salmonella typhimurium* using AuNP tags, together with differential pulse voltammetry (DPV) responses and calibration curve. Adapted from [19] with permission; (**B**) DPV curves obtained for the magnetosandwich immunoassay using AuNPs of different sizes: (a) blank, (b) 80 nm, (c) 20 nm and (d) 5 nm; and scheme of the process occurring on the electrode surface. Adapted from [23] with permission.

**Figure 2 sensors-19-05137-f002:**
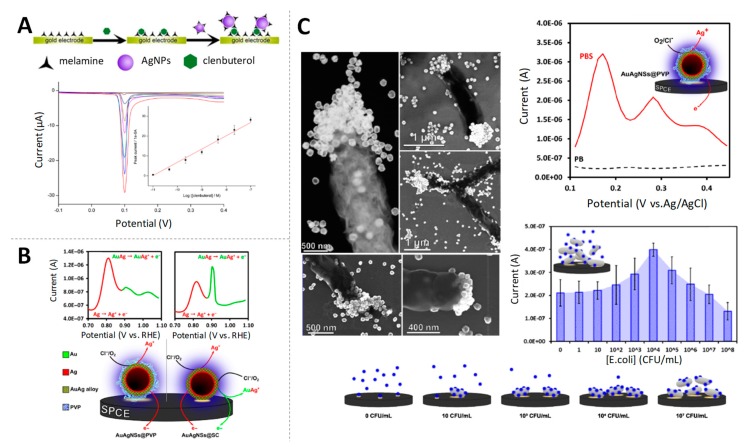
Silver nanoparticles (AgNPs) as electroactive labels (**A**) Schematic representation of the melanine functionalized AgNP-based electrochemical biosensor for the quantification of clenbuterol, linear sweep voltammetry (LSV) responses for increasing concentrations of clenbuterol and, inset, calibration curve of peak current values vs. the logarithm of clenbuterol concentration. Adapted from [26] with permission; (**B**) DPVs of AuAg NPs coated with polyvinyl pyrrolidone (PVP, left) and sodium citrate (SC, right). The analytical peak at +0.8 V corresponds to the stripping oxidation of metallic silver, while the peak at +0.9 V corresponds to the oxidation of the alloyed silver. In the bottom, proposed electrochemical mechanism for the AuAg NPs voltammetric profile. Adapted from [29] with permission; (**C**) Scanning transmission electron microscope (STEM) images of *E. coli* cells with AuAg NPs specifically linked; comparison of DPV curves of AuAgNPs in different buffers and *E. coli* detection through incubation with AuAg NPs and DPV measurements, with bacteria concentration ranging from 10^1^ to 10^8^ CFU/mL. Adapted from [30] with permission.

**Figure 3 sensors-19-05137-f003:**
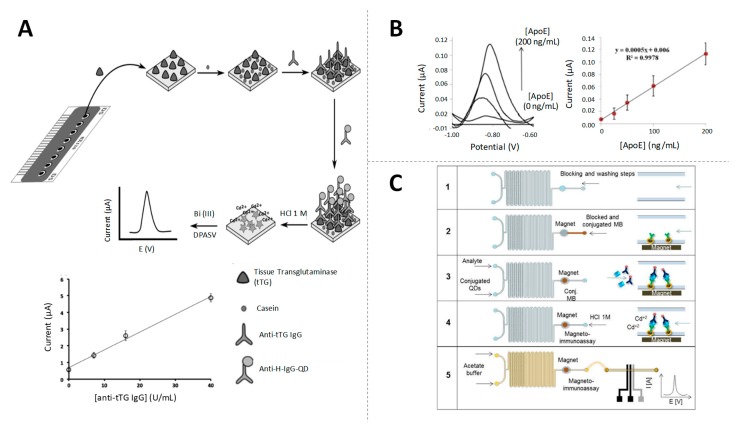
Quantum dots (QDs) as electroactive labels. (**A**) Schematic diagram of the electrochemical biosensor array for the quantification of anti-tissue transglutaminase (anti-tTG immunoglobulin G (IgG)) antibodies, based on the detection of QDs and linear response of the sensor for different concentrations of anti-tTG IgG antibody. Adapted from [41] with permission; (**B**) Performance of an ApoE-magnetoimmunoassay using QDs and calibration curve of ApoE between 0 and 200 ng/mL. Adapted from [45] with permission; (**C**) Scheme of an in-chip magnetoimmunoassay for human IgG (HIgG) detection using QDs tags. Adapted from [46] with permission.

**Figure 4 sensors-19-05137-f004:**
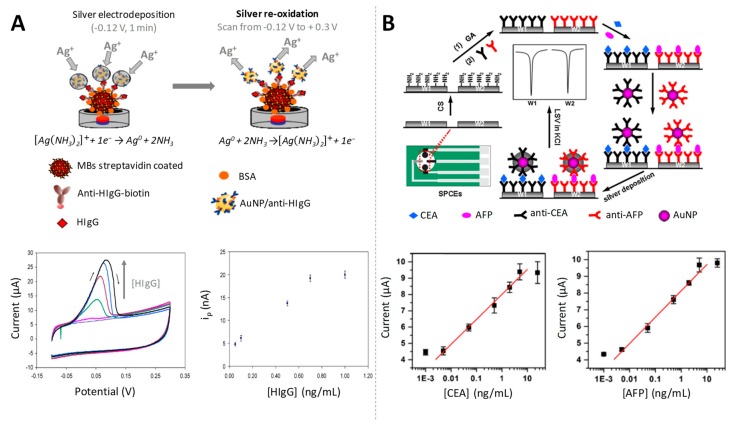
Gold nanoparticles as electrocatalysts of silver electrodeposition. (**A**) (up) Scheme of the electrochemical detection of AuNPs labels in a magnetosandwich assay for HIgG quantification based on the silver electrocatalytic deposition; (down) cyclic voltammograms (CVs) recorded in aqueous 1.0 M NH_3_ – 2.0 × 10^−4^ M AgNO_3_, from −0.12 to +0.30 V, for increasing HIgG concentrations, and relationship between the different concentrations of HIgG and the obtained peak currents used as analytical signals. Adapted from [74] with permission; (**B**) Schematic representation of the preparation of an immunosensor array and detection strategy for the detection of carcinoembryonic antigen (CEA) and alpha-fetoprotein (AFP) by LSV analysis of silver electrodeposited on AuNP tags and calibration curves for simultaneous multiplexed detection of CEA (left) and AFP (right) using the proposed strategy. Adapted from [75] with permission.

**Figure 5 sensors-19-05137-f005:**
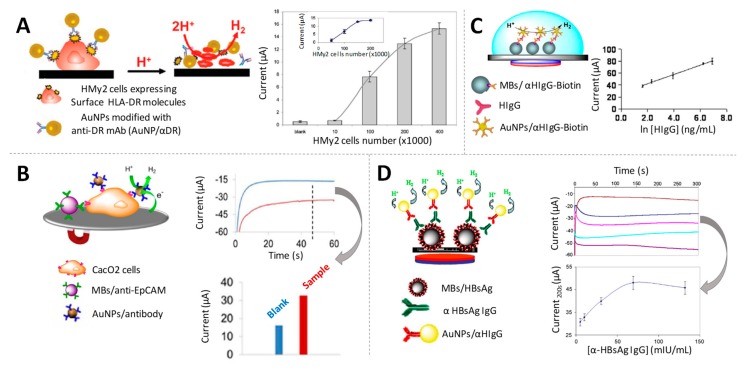
Gold nanoparticles as electrocatalysts of the hydrogen evolution reaction (HER). (**A**) (left) Scheme of an immunoassay to specifically identify HMy2 tumoral cells based on the HER electrocatalytic detection; (right) Effect of the number of HMy2 cells on the electrocatalytic signals. Adapted from [78] with permission; (**B**) (left) Scheme of the detection of Caco2 cells through the HER electrocatalysed by the AuNPs labels; (right) Chronoamperograms registered in 1 M HCl, during the HER applying a constant voltage of −1.0 V. The comparison of the corresponding analytical signals is also shown. Adapted from [80] with permission; (**C**) (left) Scheme of a magnetosandwich assay for HIgG quantification; (right) Relationship between the analytical signal and the logarithm of HIgG concentration. Adapted from [81] with permission; (**D**) (left) Scheme of the a magnetoimmunoassay for the detection of anti-hepatitis B antibodies; (right) Chronoamperograms recorded in 1M HCl for sera containing increasing concentrations of anti-hepatitis B antibodies, and effect of their concentration on the analytical signal. Adapted from [82] with permission.

**Figure 6 sensors-19-05137-f006:**
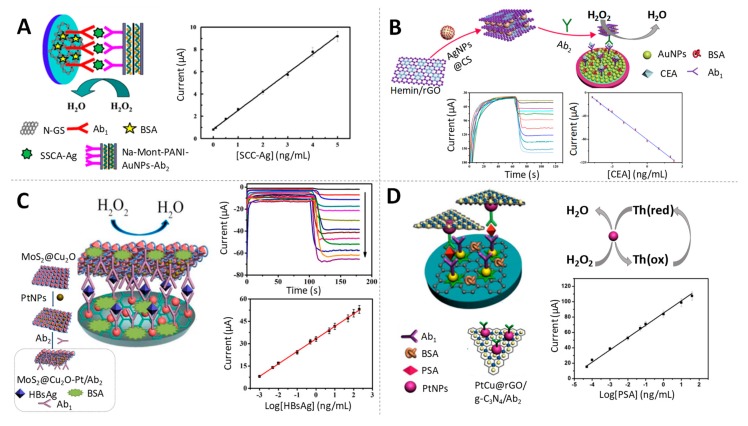
Gold, silver and platinum-based nanoparticles as electrocatalysts of H_2_O_2_ reduction. (**A**) Scheme of the immunosensor used to quantify SCC-Ag based on AuNPs and the corresponding calibration curve. Adapted from [87] with permission; (**B**) Scheme of the immunosensor used to quantify CEA based on AgNPs and the calibration curve. Adapted from [89] with permission. (**C**) Preparation procedure of MoS_2_@Cu_2_O-Pt/Ab_2_ nanocomposite and schematic representation of the immunosensor for HBsAg detection using such nanocomposite as an electrocatalytic label, together with the amperometric response of the immunosensor to different concentrations of HBsAg and the calibration curve. Adapted from [90] with permission. (**D**) Schematic representation of the immunosensor for prostate specific antigen (PSA) detection using tPtCu@rGO/g-C_3_N_4_ electrocatalytic label and calibration curve. Adapted from [91] with permission.

**Figure 7 sensors-19-05137-f007:**
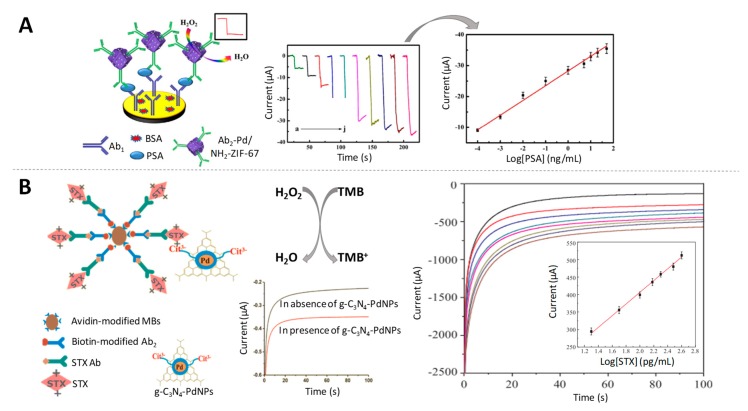
Palladium-based nanoparticles as electrocatalysts of H_2_O_2_ reduction. (**A**) Schematic representation of the immunosensor for PSA detection using Pd/NH_2_-ZIF-67 as electrocatalytic label, together with the amperometric response of the immunosensor to different concentrations of PSA and the calibration curve. Adapted from [98] with permission; (**B**) (left) Scheme of a non-competitive magnetic electrochemical immunosensor for STX quantification based on the H_2_O_2_ mediated oxidation of 3,3’,5,5’-Tetramethylbenzidine (TMB) and the current-time curves for detecting different concentration of STX, with the calibration curve as inset. Adapted from [100] with permission.

**Figure 8 sensors-19-05137-f008:**
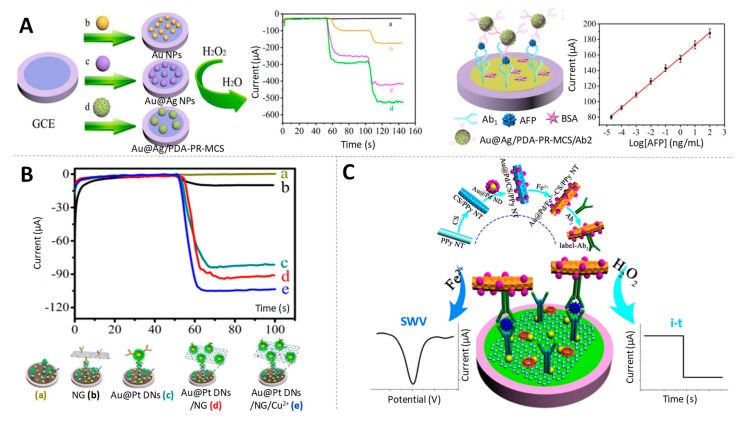
Gold-based multimetallic nanoparticles as electrocatalysts. **(A**) (left) Amperometric responses of different signal labels: bare glassy carbon electrode (GCE) (a), AuNPs (b) Au@Ag NPs (c) and core@shell gold@silver NPs loaded by polydopamine functionalized phenolic resin microporous carbon spheres (Au@Ag/PDA-PR-MCS) (d) modified GCE; (right) Schematic representation of the sandwich-type electrochemical immunosensor for AFP detection using Au@Ag/PDA-PR-MCS as electrocatalytic label. Adapted from [107] with permission; (**B**) Current responses of immunosensor for detecting 1 ng/mL CEA with different signal labels including without label (a), NG (b), Au@Pt DNs (c), Au@Pt DNs/NG (d), Au@Pt DNs/NG/Cu^2+^ (e). Adapted from [109] with permission. (**C**) Scheme of the electrochemical immunosensor for CEA quantification based on the direct and catalytic detection of Au@Pd NDs/Fe ^2+^-CS/PPy NTs. Adapted from [111] with permission.

**Figure 9 sensors-19-05137-f009:**
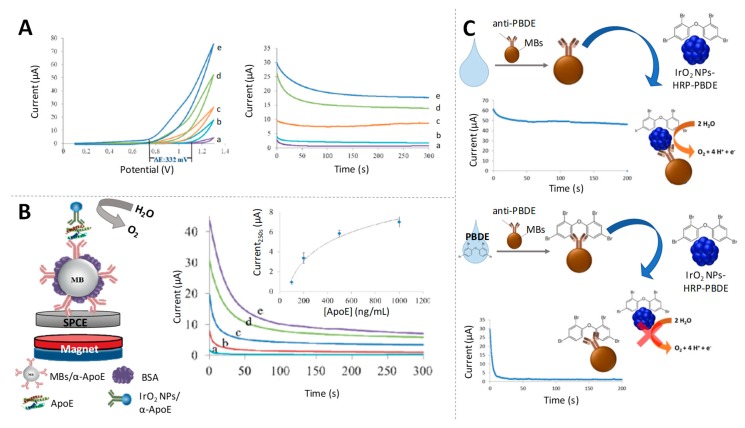
Iridium oxide nanoparticles as electrocatalysts of water oxidation reaction (WOR). (**A**) Cyclic voltammograms (left) and chronoamperograms (recorded at +1.3 V) (right) for increasing concentrations of IrO_2_ NPs in 0.1 M phosphate-buffered saline (PBS) pH 7.4. Adapted from [116] with permission; (**B**) Scheme of a magnetosandwich immunoassay using IrO_2_ NPs tags for ApoE detection based on the electrocatalysed WOR, together with chronoamperograms for samples containing different concentrations of ApoE. Calibration curve appears as inset. Adapted from [116] with permission; (**C**) Schematic representation of the competitive electrocatalytic assay for polybrominated diphenyl ether (PBDE) detection using IrO_2_ NPs as tags. Adapted from [117] with permission.

**Figure 10 sensors-19-05137-f010:**
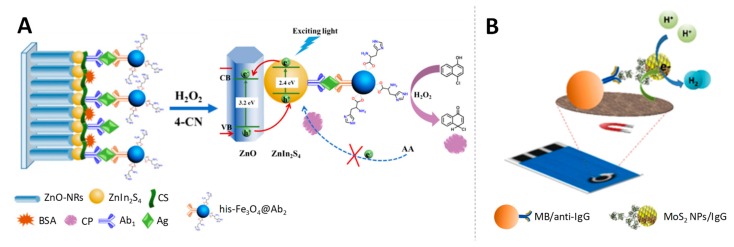
Ferromagnetic and molybdenum disulphide nanoparticles as electrocatalysts. (**A**) Development of a photoelectrochemical immunoassay using high-activity Fe_3_O_4_ nanozyme as signal amplifier and photogenerated electron-hole transfer mechanism for the detection of target antigen. Adapted from [120] with permission; (**B**) Magnetoimmunosandwich assay based on rabbit IgG-MoS_2_ NP detection through the HER mediated by the MoS_2_ label. Adapted from [122] with permission.

**Figure 11 sensors-19-05137-f011:**
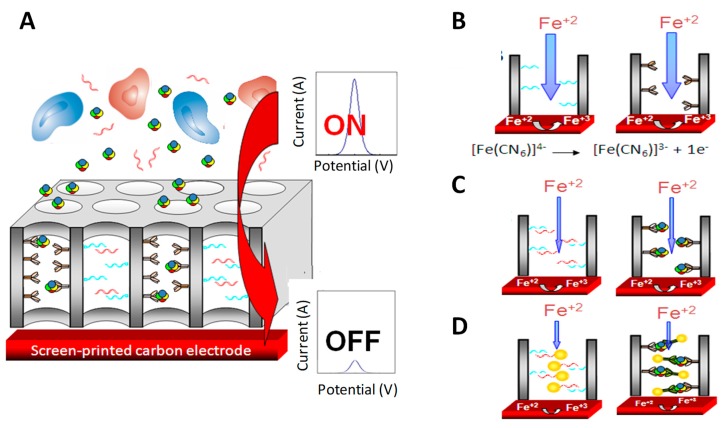
Scheme of the electrochemical sensing based on the nanoporous membranes-based platforms immunoblocking on screen-printed electrodes (SPEs). (**A**) Big molecules in the sample (i.e. blood cells) are filtered while the proteins enter inside and are captured by specific antibodies; (**B**) Sensing principle in the absence and (**C**) in the presence of the specific protein (or target single strand DNA (ssDNA)) in the sample, and (**D**) in the case of the assay performed using AuNP tags as blocking agents. Adapted from [125] with permission.

**Figure 12 sensors-19-05137-f012:**
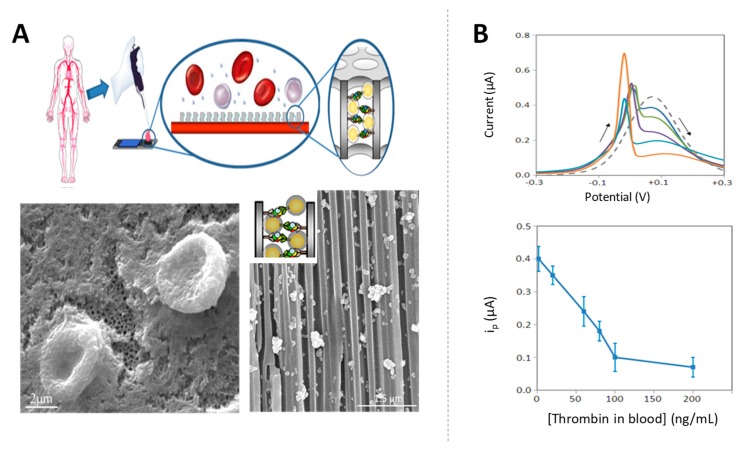
Signal enhancement in nanochannels by AuNP tags and silver deposition: application to biomarkers detection in blood. (**A**) Scheme of the sensing strategy together with scanning electron microscope (SEM) images of the alumina nanoporous membrane corresponding to (left) a top view of a membrane on which a drop of blood was deposited (red blood cells remain out of the membrane); (right) a cross-sectional view of a membrane with immobilized anti-thrombin antibodies, left to react with a blood sample containing thrombin biomarker and completing the sandwich using AuNP tags and silver enhancement (silver crystals blocking the nanochannels are observed); (**B**) DPV responses registered for blood samples containing different concentrations of thrombin and effect of the concentration of thrombin on the voltammetric peak current. Adapted from [130] with permission.

**Table 1 sensors-19-05137-t001:** Electrochemical immunosensors reported over the last five years, involving the use of nanoparticles as labels in electrochemical immunosensing.

NPs	Strategy	Analyte	LOD	Reference
AuNPs	Direct detection	*Salmonella typhimurium*	7.7 cells/mL	[19]
AuNPs	Direct detection	CEA AFP	4.6 pg/mL 3.1 pg/mL	[20]
AgNPs	Direct detection	Clenbuterol	3.14 pg/mL	[26]
AgNPs	Direct detection	TBEV	50 IU/mL	[27]
AuAgNPs	Direct detection	*E. coli*	10^2^ CFU/mL	[30]
CdSe@ZnS QDs	Direct detection	anti-tTG IgG	2.2 U/mL	[41]
CdSe@ZnS QDs	Direct detection	anti-tTG IgA	2.4 U/mL	[42]
CdSe@ZnS QDs	Direct detection	ApoE	12.5 ng/mL	[45]
CdSe@ZnS QDs	Direct detection	HIgG	3.5 ng/mL	[46]
SA/Au/CNH	Silver electrodeposition-based detection	CEA AFP	32 fg/mL 24 fg/mL	[76]
AuNPs	HER-based detection	ApoE Beta amyloid	80 pg/mL 19 pg/mL	[83]
AuNPs	HER-based detection	*E. coli* O157:H7	457 CFU/mL (in minced beef) 309 CFU/mL (in tap water)	[84]
Na-Mont-PANI-AuNPs	H_2_O_2_ reduction-based detection	SCC-Ag	300 fg/mL	[87]
PS@PEI@CAT@AuNPs	H_2_O_2_ reduction-based detection	TBBPA-DHEE	120 pg/mL	[88]
AgNPs@CS-Hemin/rGO	H_2_O_2_ reduction-based detection	CEA	6.7 fg/mL	[89]
MoS_2_@Cu_2_O-PtNPs	H_2_O_2_ reduction-based detection	HBsAg	150 fg/mL	[90]
PtCu@rGO/g-C_3_N_4_	H_2_O_2_ reduction-based detection	PSA	16.6 fg/mL	[91]
Ag@Au-Fe_3_O_4_	H_2_O_2_ reduction-based detection	HIgG	50 fg/mL	[92]
Rh@Pt NDs/NH_2_-GS	H_2_O_2_ reduction-based detection	HBsAg	166 fg/mL	[95]
PtNPs/Co_3_O_4_/graphene	H_2_O_2_ reduction-based detection	AFP	29 fg/mL	[96]
Pd/APTES-M-CeO_2_-GS	H_2_O_2_ reduction-based detection	AFP	33 fg/mL	[97]
Pd/NH_2_-ZIF-67	H_2_O_2_ reduction-based detection	PSA	30 fg/mL	[98]
g-C_3_N_4_-PdNPs	H_2_O_2_ reduction-based detection	STX	1.2 pg/mL	[100]
Au@Ag/Au core@double shell NPs	H_2_O_2_ reduction-based detection	SCC-Ag	180 fg/mL	[104]
Co_3_O_4_@CeO_2_-Au@Pt	H_2_O_2_ reduction-based detection	SCC-Ag	33 fg/mL	[105]
Au@Ag-Cu_2_O	H_2_O_2_ reduction-based detection	PSA	3 fg/mL	[106]
Au@Ag/PDA-PR-MCS	H_2_O_2_ reduction-based detection	AFP	6.7 fg/mL	[107]
rGO-NR Au@Pt	H_2_O_2_ reduction-based detection	*E. coli* O157:H7	91 CFU/mL	[108]
Au@Pt DNs/NG/ Cu^2+^	H_2_O_2_ reduction-based detection	CEA	167 fg/mL	[109]
Au@Pd/MoS_2_@MWCNTs	H_2_O_2_ reduction-based detection	HBeAg	26 fg/mL	[110]
Au@Pd NDs/Fe^2+^-CS/PPy NTs	Direct detection H_2_O_2_ reduction-based detection	CEA	167 fg/mL 17 fg/mL	[111]
M-Pd@Pt/NH_2_-GS	H_2_O_2_ reduction-based detection	PSA	3.3 fg/mL	[112]
m-PdPtCu	H_2_O_2_ reduction-based detection	PSA	3.3 fg/mL	[113]
IrNPs	H_2_O_2_ reduction-based detection	CEA	230 fg/mL	[114]
IrO_2_ NPs	Water oxidation reaction-based detection	ApoE	68 ng/mL	[116]
IrO_2_ NPs	Water oxidation reaction-based detection	PBDE	21.5 ppb	[117]
his-Fe_3_O_4_	H_2_O_2_ reduction-based detection	PSA	18 fg/mL	[120]
MoS_2_ NPs	Hydrogen evolution reaction-based detection	Rabbit IgG	1.94 pg/mL	[122]
AuNPs	Nanochannel blocking agent	HIgG	580 ng/mL	[127]
AuNPs	Nanochannel blocking agent	PTHrP	50 ng/mL	[128]

AFP: Alpha-fetoprotein; Ag@Au-Fe_3_O_4_: Nanocomposite made of Fe_3_O_4_ nanospheres and Ag@Au nanorods; AgNPs: Silver nanoparticles; AgNPs@CS-Hemin/rGO: Microporous carbon spheres loading silver NPs spaced Hemin/reduced graphene oxide porous nanocomposite; anti-tTG IgA: Anti-tissue transglutaminase immunoglobulin A antibodies; anti-tTG IgG: Anti-tissue transglutaminase immunoglobulin G antibodies; ApoE: Apolipoprotein E; AuNPs: Gold nanoparticles; Au@Ag-Cu_2_O: Gold@silver core@shell NPs and disordered cuprous oxide nanocomposite; Au@Ag/PDA-PR-MCS: Core@shell gold@silver NPs loaded by polydopamine functionalized phenolic resin microporous carbon spheres nanocomposite; Au@Pd/MoS_2_@MWCNTs: Gold@palladium NPs loaded with molybdenum disulfide functionalized multiwalled carbon nanotubes nanocomposite; Au@Pd NDs/Fe^2+^-CS/PPy NTs: Gold@palladium nanodendrites loaded on ferrous-chitosan functionalized polypyrrole nanotubes nanocomposite; Au@Pt DNs/NG/ Cu^2+^: Au@Pt NPs dendritic nanomaterial functionalized with nitrogen-doped graphene loaded with copper ions nanocomposite; CdSe@ZnS QDs: CdSe@ZnS quantum dots; Co_3_O_4_@CeO_2_-Au@Pt: Amino functionalized cobaltosic oxide@ceric dioxide nanocubes combined with gold@platinum NPs; CEA: Carcinoembryonic antigen; *E. coli*: *Escherichia coli*; g-C_3_N_4_-PdNPs: Palladium-doped graphitic carbon nitride nanosheets; HBeAg: Hepatitis B e antigen; HBsAg: Hepatitis B surface antigen; HIgG: Human IgG; his-Fe_3_O_4:_ Histidine-modified Fe_3_O_4_ NPs; IrNPs: Iridium nanoparticles; IrO_2_ NPs: Iridium oxide nanoparticles; M-Pd@Pt/NH_2_-GS: Mesoporous core@shell palladium@platinum NP loaded by amino group functionalized graphene nanocomposite; m-PdPtCu: Trimetallic PdPtCu nanospheres; MoS_2_ NPs: Molybdenum disulphide NPs; MoS_2_@Cu_2_O-PtNPs: Molybdenum disulphide@cuprous oxide–platinum nanocomposite; Na-Mont-PANI-AuNPs: Na-montmorillonite-polyaniline-AuNPs nanocomposite; PBDE: Polybrominated diphenyl ether; Pd/APTES-M-CeO_2_-GS: Graphene oxide and CeO_2_ mesoporous nanocomposite functionalized by 3-aminopropyltriethoxysilane supported Pd octahedral NPs; Pd/NH_2_-ZIF-67: Pd NPs loaded electroactive amino-zeolitic imidazolate framework-67 nanocomposite; PSA: Prostate specific antigen; PS@PEI@CAT@AuNPs: Catalase functionalized AuNPs-loaded self-assembled polymer nanospheres; PtCu@rGO/g-C_3_N_4_: PtCu bimetallic hybrid loaded on 2D/2D reduced graphene oxide/graphitic carbon nitride nanocomposite; PTHrP: Parathyroid hormone-related protein; PtNPs/Co_3_O_4_/graphene: PtNPs anchored on cobalt oxide/graphene nanosheets; rGO-NR Au@Pt: Gold@platinum core@shell NPs modified with neutral red functionalized reduced graphene oxide nanocomposite; Rh@Pt NDs/NH_2_-GS: Nanocomposite based on Rh core and Pt shell nanodendrites loaded onto functionalized graphene nanosheets; SA/Au/CNH: Streptavidin/nanogold/ carbon nanohorn; SCC-Ag: Squamous cell carcinoma antigen; STX: Saxitoxin; TBBPA-DHEE: Tetrabromobisphenol A bis(2-hydroxyethyl) ether; TBEV: Tick-borne encephalitis virus.

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
