# Peer review of "Nanoparticles as Emerging Labels in Electrochemical Immunosensors"

_sensors, 2019, doi:10.3390/s19235137_

Round 1
Reviewer 1 Report
This review paper presents an overview about relevant and recent applications of nanoparticles as labels in electrochemical immunosensors. I think it is very well written and present some very interesting work reported in the literature. The critique on reported works is also appropriated, thus my recommendation is to publish as it is.
Author Response
We thank very much the referee for his/her comments.
Reviewer 2 Report
The manuscript entitled “nanoparticles as emerging labels in electrochemical immunosensors” has summarized three different kinds of applications about nanoparticles as labels in electrochemical immunosensors, and it provides comprehensive description of detailed nanoparticles, I therefore recommend the publication of this manuscript in this journal after revision: 1. Is there any other review about nanoparticles as emerging labels in electrochemical immunosensors? The main intention or advantage of this review is not mentioned in the manuscript. I suggest the authors adding one or two sentences on the advantages of the review in abstract or introduction. 2. Graphical abstract should be added in the revised manuscript to illustrate the subsections of this review. 3. Some related references are suggested to be cited into this paper, such as Xu et al., Biosensors and Bioelectronics 56 (2014) 174–179; Zhou et al., Analytica Chimica Acta 969 (2017) 8–17; Zhao et al., Analytica Chimica Acta 847 (2014) 37–43. 4. English presentation needs to be improved.
Author Response
Question 1: Is there any other review about nanoparticles as emerging labels in electrochemical immunosensors? The main intention or advantage of this review is not mentioned in the manuscript. I suggest the authors adding one or two sentences on the advantages of the review in abstract or introduction.
Answer: We thank the referee for bringing up this point. Despite the increasing interest and number of published works on NPs as labels in electrochemical biosensing, there is a lack of recent reviews covering this field, mainly focusing on immunosensing. A sentence highlighting this issue is now included at the introduction section of the revised manuscript.
Question 2: Graphical abstract should be added in the revised manuscript to illustrate the subsections of this review.
Answer: We agree with the referee’s comment. A graphical abstract illustrating the different topics covered throughout this review is now included in the revised manuscript.
Question 3: Some related references are suggested to be cited into this paper, such as Xu et al., Biosensors and Bioelectronics 56 (2014) 174–179; Zhou et al., Analytica Chimica Acta 969 (2017) 8–17; Zhao et al., Analytica Chimica Acta 847 (2014) 37–43.
Answer: In agreement with the referee’s comment, the suggested references are now included in the revised manuscript.
Question 4: English presentation needs to be improved.
Answer: In agreement with the referee’s comment, English has been revised and some grammatical and typographic errors have been corrected.
Reviewer 3 Report
This interesting review paper is devoted to specific labeling applications of nanoparticles (NPs) in electrochemical immunosensors. The scientific quality of the manuscript and its importance for the field of electrochemistry and bioanalytical chemistry are undoubted. I really appreciate the amount of work performed for this manuscript preparation. There are only few points which could be improved during revision of the manuscript:
In the manuscript, following papers closely related to the investigated topic could also be cited:
Khristunova Y., Korotkova E., Kratochvil B., Barek J., Dorozhko E., Vyskocil V., Plotnikov E., Voronova O., Sidelnikov V.: Preparation and Investigation of Silver Nanoparticle–Antibody Bioconjugates for Electrochemical Immunoassay of Tick-Borne Encephalitis. Sensors 2019, 19, 2103.
Fojta M., Daňhel A., Havran L., Vyskočil V.: Recent Progress in Electrochemical Sensors and Assays for DNA Damage and Repair. TrAC Trends in Analytical Chemistry 2016, 79, 160–167.
The term “red-ox” should be replaced by the classical one “redox”. In the Introduction, some general/tutorial information about various roles of NPs in nanoparticle-based immunosensors should be implemented. The reader should have at the beginning a general information how the NPs can be utilized in this field. What about carbon-based nanoparticles. Are they also applied in immunosensing strategies? It would be nice to include a survey table showing a list of analytes, strategies, and obtained results (e.g., LOD values) published in the field of NP-based immunosensors for some recent period of time (1, 3, or 5 years, depending on number of these papers). Some references in the List of References are not complete, missing some important data to (easily) find them. It would be nice to check them. A few text imperfections (typos) should be corrected during a careful proof-reading.
Author Response
Question 1: In the manuscript, following papers closely related to the investigated topic could also be cited: Khristunova Y., Korotkova E., Kratochvil B., Barek J., Dorozhko E., Vyskocil V., Plotnikov E., Voronova O., Sidelnikov V.: Preparation and Investigation of Silver Nanoparticle–Antibody Bioconjugates for Electrochemical Immunoassay of Tick-Borne Encephalitis. Sensors 2019, 19, 2103. Fojta M., Daňhel A., Havran L., Vyskočil V.: Recent Progress in Electrochemical Sensors and Assays for DNA Damage and Repair. TrAC Trends in Analytical Chemistry 2016, 79, 160–167.
Answer: In agreement with the referee’s comment, the suggested references are now included in the revised manuscript.
Question 2: The term “red-ox” should be replaced by the classical one “redox”. In the Introduction, some general/tutorial information about various roles of NPs in nanoparticle-based immunosensors should be implemented. The reader should have at the beginning a general information how the NPs can be utilized in this field.
Answer: We thank the referee for bringing up these points. The term “red-ox” has been replaced by “redox” in the revised manuscript. General information about the roles of NPs in immunosensors (mostly as modifiers of electrode surfaces) is also now briefly given at the introduction section of the revised manuscript.
Question 3: What about carbon-based nanoparticles. Are they also applied in immunosensing strategies?
Answer: We thank the referee for bringing up this point. Carbon-based nanoparticles have been applied in electrochemical immunosensing but mostly as modifiers of electrode surfaces, so such applications are out of the scope of this review article. Anyway, such use of carbon-based materials together with some representative references is now included at the introduction section of the revised manuscript.
Question 4: It would be nice to include a survey table showing a list of analytes, strategies, and obtained results (e.g., LOD values) published in the field of NP-based immunosensors for some recent period of time (1, 3, or 5 years, depending on number of these papers).
Answer: In agreement with the referee’s suggestion, a table summarizing analytes, strategies and results is now given at the revised manuscript.
Question 5: Some references in the List of References are not complete, missing some important data to (easily) find them. It would be nice to check them. A few text imperfections (typos) should be corrected during a careful proof-reading.
Answer: We thank the referee for noticing such issue. The list of references has been carefully revised and the typographic errors have been corrected.